# Reconstruction of Simulated Cylindrical Defects in Acrylic Glass Plate Using Pulsed Phase Thermography

**Ljubiša Tomić** [1] , **Vesna Damnjanović** [2,*], **Goran Dikić** [3] **and Bojan Milanović** [3]

1    Military Technical Institute, Department of electronics, Ratka Resanovica 1, Belgrade 11030, Serbia; ljubisa.tomic@gmail.com
2    Faculty of Mining and Geology, University of Belgrade, Djušina 7 Belgrade 11000, Serbia
3    Military Academy, University of Defence, Pavla Jurišića Šturma 33 Belgrade 11000, Serbia; dikic.goran@gmail.com (G.D.); bojan.milanovic@va.mod.gov.rs (B.M.)
*    Correspondence: vesna.damnjanovic@rgf.bg.ac.rs; Tel.: +381-11-3219109

**Abstract:** The results of testing of acrylic glass, in which cylindrical defects were simulated at different depths by applying Pulsed Phase Thermography, are presented in the paper. To ensure better visibility of the simulated defects, suitable thermal images were selected and then processed by using two different procedures. In the first procedure, reduced thermal image sequences were generated by uniform extraction from the basic sequence, to enable analysis at different sampling frequencies. The second procedure was based on the application of a window function, which ensured that only uniformly selected thermal images took part in the evaluation of the basic sequence. The remaining thermal images were not used, but they did participate in the determination of the length of the analyzed sequence; in other words, their existence was registered through the number of samples used in Fast Fourier Transformation. The second procedure yielded much better results with regard to the estimation of the shape of a defect and the depth at which it was located. To provide better insight into the development of the thermal process in the defect area, an additional analysis of pixel intensity variation in the time domain was undertaken.

**Keywords:** Nondestructive testing; Pulsed Phase Thermography; acrylic glass; cylindrical defects; Fast Fourier Transformation

## 1. Introduction

Infrared (IR) Thermography is a widely used non-destructive method for quick testing and evaluation of materials and structures (Non-destructive Testing & Evaluation—NDT&E) [1–6]. Depending on how the tested material is being heated, IR Thermography is either passive or active [7]. The most popular active IR thermography methods are Pulsed Thermography (PT), Lock-in Thermography (LT) and Pulsed Phase Thermography (PPT) [8–14]. PPT in fact integrates PT and LT, as it is quick and simple, like PT, and provides phase delay information, like LT. It involves Discrete Fourier Transformation (DFT). An optimized version is generally used in practice, known as Fast Fourier Transformation (FFT) [15]. FFT produces results in the form of ampligrams and phasegrams which are, in contrast to thermal images, less affected by ambient reflection [10]. It should be noted that phasegrams are virtually insensitive to non-uniform heating so that, compared to amplitude-based methods, they enable defect detection at greater depths [12].

The application of the PT method to test defects in acrylic glass, which is extensively used at present, is coupled with problems during acquisition and analysis of captured thermal images, given that the surface of the material is highly reflective and there are issues associated with homogeneous

heating and longer heat diffusion. Defect depths can be estimated by processing a sequence of thermal images captured at different sampling frequencies. Problems arise where the lowest sampling frequency that a specific camera supports is higher than required. The challenge is how to attain, through data processing, the effects achievable with a camera that supports lower frequencies as well. In the present research, two different procedures are followed to process sequences of thermal images, in order to compare their applicability to cases that involve the detection of simulated defects and determination of their sizes and depths, in acrylic test samples [16,17]. In the first procedure, the effect of capturing at a lower sampling frequency was simulated by uniform frame extraction from the available sequence. This resulted in a smaller number of frames, so that FFT processing provided fewer spectral components. However, the visibility of the defects in the tested material decreased and for that reason, in the second procedure the number of available frames was not reduced, but a window function in the time domain enabled only the frames extracted in the previous procedure to take part in the calculations of spectral components, while disregarding the effect of all the remaining frames.

## 2. Experimental Setup and Test Sample

The experimental setup was comprised of heat sources, a thermal camera and a computer that recorded data in real time. The heat sources were two photographic flashes (BOWENS BW-3955 Gemini R & Pro). The duration of the heat pulse at a maximum power of 1500 W was 0.7142 ms. Homogeneous heating of the test sample was achieved by positioning the flashes at an optimal distance of about 35 cm, at an angle of 45° relative to the normal of the surface being heated. To better direct the heat flux to the test sample surface, 65° reflectors with a 20 cm aperture were mounted on the flashes. Triggering of the flashes was synchronized via the computer. The cooling process was monitored by using a thermal camera (FLIR SC620), designed for the spectral range 7.5 μm–13 μm. The camera can be operated in three modes, so that the duration of a frame can be 8.33 ms, 16.66 ms or 33.33 ms.

Figure 1 (photograph) shows the arrangement of cylindrical defects (flat bottom holes) simulated in a black acrylic glass plate. The dimensions of the plate are 180 mm × 50 mm × 4.2 mm. Four series of defects were made and denoted by A, B, C and D. Series A comprised four holes of equal diameters (∅ 5 mm) but different depths $h$ (2.7 mm, 3.2 mm, 3.4 mm and 3.7 mm, respectively). Depths $h$ are distances from the damaged surface. Series B, C, D and E included three holes each, whose diameters differed (∅ 3 mm, ∅ 8 mm and ∅ 15 mm, respectively), but their depth was the same. The depth of series B was $h = 3.9$ mm, of C $h = 3.5$ mm, of D $h = 3.2$ mm, and of E $h = 2.6$ mm.

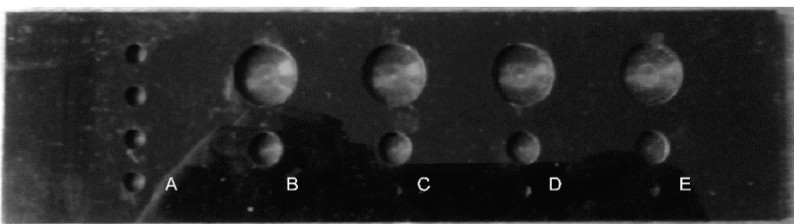

**Figure 1.** Photograph of acrylic glass test sample side on which defects were simulated marked as A, B, C, D and E series.

The dimensions of the defects are shown in Table 1, where ∅ is the cylindrical hole diameter and the other dimensions $h$ are the corresponding depths. The distance between the defects in group A was 10 mm and the distance from the test sample edge to the largest, medium-size and smallest defects in the other groups of defects was 15 mm, 32.5 mm and 42.5 mm, respectively.

A series of measurements were made after the smooth, defect-free side of the test sample was illuminated.

**Table 1.** Dimensions of defects simulated in 4.2 mm thick acrylic glass sample.

| A Ø 5 mm | B | C | D | E |
|---|---|---|---|---|
| $h$ = 2.7 mm | $h$ = 3.9 mm | $h$ = 3.5 mm | $h$ = 3.2 mm | $h$ = 2.6 mm |
| $h$ = 3.2 mm | Ø 3 mm | Ø 3 mm | Ø 3 mm | Ø 3 mm |
| $h$ = 3.4 mm | Ø 8 mm | Ø 8 mm | Ø 8 mm | Ø 8 mm |
| $h$ = 3.7 mm | Ø 15 mm | Ø 15 mm | Ø 15 mm | Ø 15 mm |

## 3. Results and Discussion

Data processing, based on the DFT algorithm, allowed for the sequence of samples of signal $x(k)$, recorded in the time domain, to be described by a set of spectral components $X(n)$ in the frequency domain. The number of spectral components in the frequency domain was equal to the number of samples $N$ in the time domain, where the selected number of samples $N$ was the integer power of 2.

Given that DFT, which is defined by Equation (1), yielded results in complex form, the temperature profile of each pixel in the sequence of recorded thermal images could be described by the corresponding amplitude and phase characteristic in the frequency domain. These characteristics were used to generate corresponding amplitude (PT method) and phase (PPT method) maps, which made the defects more easily detectable than in the case of analysis of the original thermal image.

$$X(n) = \sum_{k=0}^{N-1} x(k)e^{-i\frac{2\pi}{N}nk} \quad n = 0, 1, \ldots, N-1. \tag{1}$$

To assess the depth of a defect, it is necessary to process data recorded at different sampling frequencies. In the specific case, this was achieved by pre-programmed extraction of suitable frames from the basic sequence recorded at a camera sampling frequency of $f_s$ = 30 frames per second. Two procedures were followed.

In the first procedure (Procedure A), 64 frames were extracted in the order: $k$ = 3, 6, 9, 12, 15, 18, 21 and 24, using the loop FOR $i = k_0{:}k{:}(k_0 + k{\cdot}63)$. This resulted in the minimum sampling frequency of $f_{\min}$ = 30/24 samples per second. For each pixel selected in this manner with 64 frames, vector $x(k)$ was generated and used in the DFT described by Equation (1). A total of 1617 frames were captured during the experiment. Keeping in mind that at $k$ = 24 at least $1 + 24 \times 63 = 1513$ are required, any further frequency reduction was not possible. On the other hand, if only 32 frames were used, a smaller number of spectral components would be obtained (i.e., the resolution in the frequency domain would be poorer).

To enhance resolution, in the second procedure (Procedure B) the number of samples $N$ was increased, but without adding information by selecting additional frames from the basic sequence of all recorded frames. Instead, "zero frames" were introduced (i.e., the values of all pixels in the corresponding matrix were equal to zero). This meant that a sample vector was generated each time and its every $k$-th term, beginning with the first, had the value of the corresponding pixel within the sequence of the 64 frames selected from the basic sequence, as in the case of Procedure A. All other terms were zero. The total number of terms $N$ of the vector generated in this way was equal to the minimum value of the power of 2, which ensured that the pixel values of all frames extracted from the basic sequence by means of the previously-mentioned FOR NEXT loop were stored.

It follows from Equation (1) that all samples of signal $x(k)$, recorded in the time domain, participate in the computation of all spectral components $X(n)$ in the frequency domain. The introduction of zero frames had the indirect effect of sampling frequency reduction. Namely, the frequency of appearing frames that participate with their pixel values in the computation of spectral components $X(n)$ corresponds to the desirable sampling frequency. For example, at $k$ = 3 the frequency of appearing frames whose elements differed from zero was lower by a factor of three.

In order to facilitate assessment of the changes upon heating of the test sample, the variation in intensity in areas whose size was 16 × 16 pixels, denoted by B, C, D, E and Sa in Figure 2, were

analyzed. The centers of areas B, C, D and E coincided with the centers of the cylindrical defects with the largest diameter ($\emptyset = 15$ mm). As shown, there were no defects in the sound area (Sa).

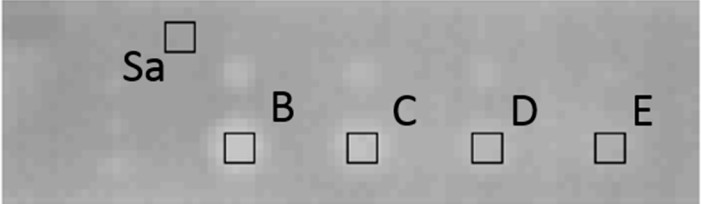

**Figure 2.** Thermal image of acrylic glass test sample: 450th frame after a light pulse.

Figure 3 shows the curves obtained by computing the mathematical expectation of pixel intensity in the said areas. To better assess the variation in intensity upon test sample irradiation, the number of frames is shown in logarithmic scale. Figure 3 clearly indicates a sudden increase in intensity when the light pulse was emitted. Immediately thereafter, the curves did not change their value due to thermal image saturation [17]. The declining values of these curves were a result of test sample cooling and marked the beginning of the thermal image sequence that could be used to detect the defects. The end of this sequence was determined by the thermal image whose change in intensity was negligible. In the specific case, the 24th frame was selected as the first ($k_0$) of 64 frames used in data processing applying the PPT algorithm.

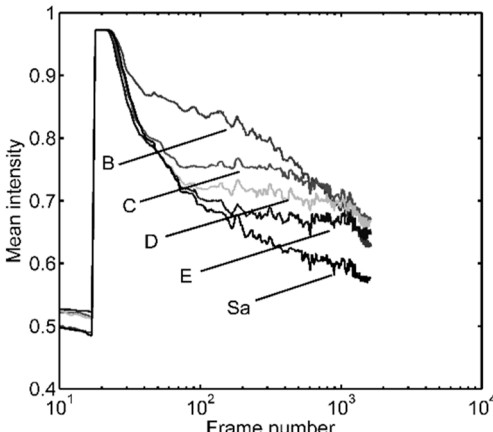

**Figure 3.** Change in mathematical expectation of pixel intensity within the areas denoted by B, C, D, E and Sa in Figure 2.

It is apparent that the cooling process began in the same way in all the areas. However, after a brief transition process, first curve B began to separate, followed by C, D and E, in that order. The curve denoted by Sa illustrates the cooling process of the sound area. The observed changes correlated with the nature of the simulated defects. Given that their diameters were the same, it was reasonable to expect that the shallowest defect (curve B) would be the first to start cooling. As a result, the cooling process in this case was the quickest (curve B declined the most rapidly). After a sufficiently long time from the beginning of cooling, all curves converged to the same value. This was not visible, of course, as there were not enough recorded frames.

Figure 4 shows the change in the mathematical expectation of pixel intensity in areas B, C, D and E, relative to Sa. The occurrence of extremes as the curves changed was in correlation with defect depth, in terms of both value and order of occurrence. First, the most pronounced maximum was noted on curve a, which corresponds to the shallowest defect ($\emptyset = 15$ mm along vertical B). The last maximum occurred at the very end of curve d and had the lowest value, as it was related to the deepest defect ($\emptyset = 15$ mm along vertical E).

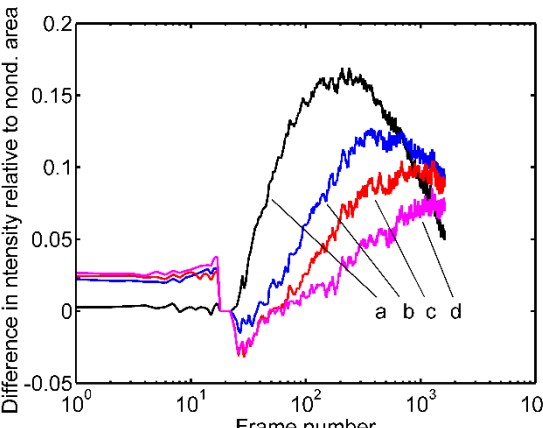

**Figure 4.** Deviation of values on curves B, C, D and E shown in Figure 3 from those of curve Sa: (a) B-Sa, (b) C-Sa, (c) D-Sa and (d) E-Sa.

Figures 5 and 6 show the results of data processing based on the FFT algorithm. The left half of these figures depicts the phase contrast obtained by processing the data extracted from the main sequence according to Procedure A. The shapes of the detected defects clearly indicate that there was a resolution problem. The results of data processing following Procedure B are shown in the right half of the figures, where the correlation of the shapes between detected and simulated defects is apparent. This is consistent with the fact that the total number of spectral components $X(n)$ in Equation (1) is determined by the number of samples $N$. A greater number yields more spectral components; this is manifested by better resolution in the figures generated applying the PPT algorithm. Of course, the number of needed computations increases, but it should be kept in mind that all the computations related to the fictitious "zero frame" pixels are left out.

In addition to enhanced resolution, a more distinct continuity of change in pixel intensity is apparent, which corresponds to the defects (especially those with the largest diameter). This ensured better insight into the thermal process dynamics and facilitated assessment of relative depth ratios.

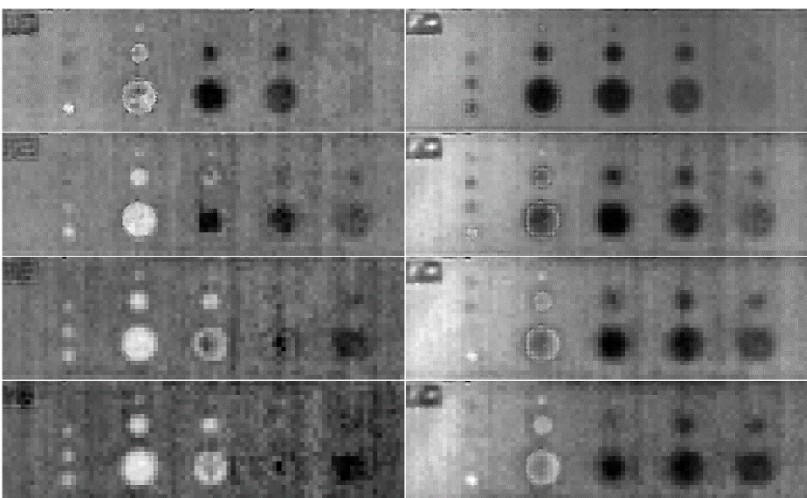

**Figure 5.** Defect phase contrast obtained by processing every 3rd, 6th, 9th and 12th frame (top down): Procedure A left, Procedure B right.

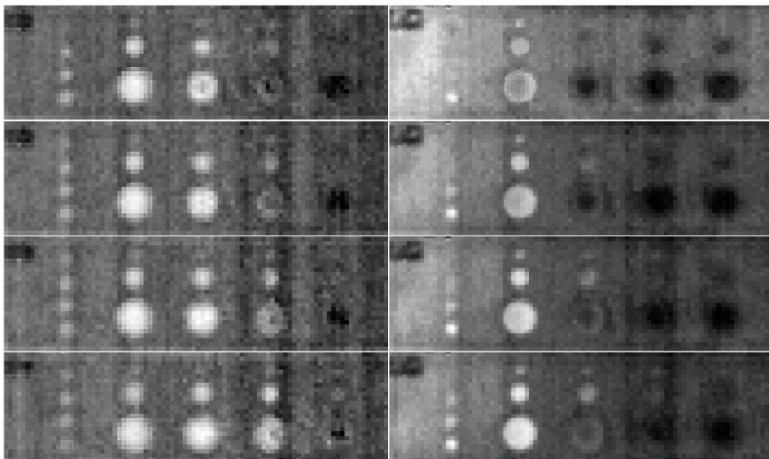

**Figure 6.** Defect phase contrast obtained by processing every 15th, 18th, 21st and 24th frame (top down): Procedure A left, Procedure B right.

Defects can be detected at a certain (specific) depth based on the phase contrast in the frequency domain, but only after the sampling frequency drops to below the blind frequency [18]. The limiting frequency is higher if the defect is shallower. It is apparent that the defects along vertical E became visible only after the sampling frequency was low enough, or when the value of step *k* in the FOR NEXT loop became greater than 3 in data processing.

It is apparent that the time lag of the peak of the corresponding curve, in Figure 4, is the longest in the case of the defects along vertical E. Namely, it corresponds to the defect that is most distant from the heated surface. Obviously, there is a correlation with the results shown in Figs. 4 and 5. As the distance of the defects (of the same radii) increases, viewed relative to the heated surface, so does the time lag. This is consistent with the fact that a longer sampling period, or a lower blind frequency, is required for them to be visible.

The relative phase contrast ratios, shown as differences in gray scale in Figures 5 and 6, were fully correlated with the detected defect depth. The shallowest defects were the lightest, as the phase contrast, determined by the temperature change after test sample irradiation, was the highest there [18]. A quantitative description of this is given in Figure 7 using curves that represent the change in the mathematical expectation of the gray scale in Figures 5 and 6, within areas whose boundaries coincide with areas B, C, D and E in Figure 2.

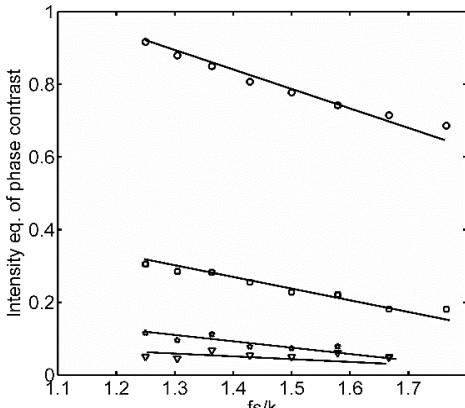

**Figure 7.** Intensity equivalent of phase contrast versus $f_s/k$ ratio for $\emptyset = 15$ mm defects: circles, squares, pentagrams and triangles along vertical B, C, D and E, respectively.

In order to better illustrate the thermal process within the largest-diameter defects, Figures 8–10 show the change in intensity of the pixels positioned in the interval from 101 to 140, within the broken

lines shown in Figure 11, whose vertical coordinates correspond to the values 62, 67, 73, 80, 83 and 87. The change in pixel intensity during the first 200 frames is shown in color, where blue is the initial, minimum value and dark red is the maximum intensity immediately after test sample irradiation.

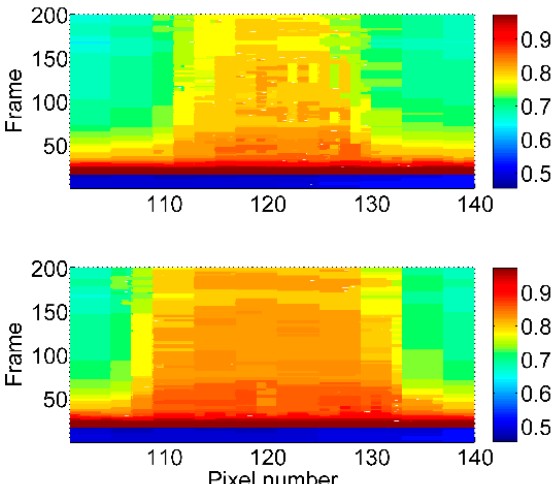

**Figure 8.** Pixel intensity variation along lines 62 (top) and 67 (bottom).

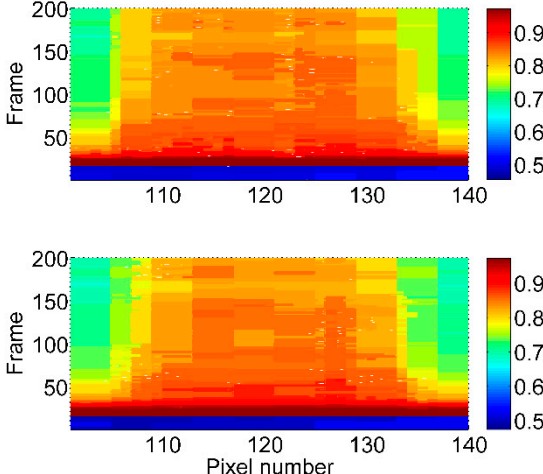

**Figure 9.** Pixel intensity variation along lines 73 (top) and 80 (bottom).

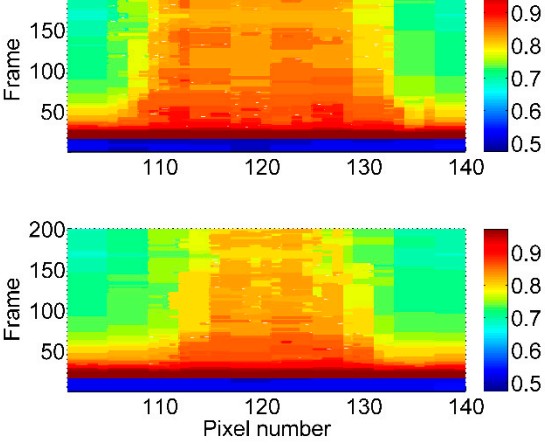

**Figure 10.** Pixel intensity variation along lines 82 (top) and 87 (bottom).

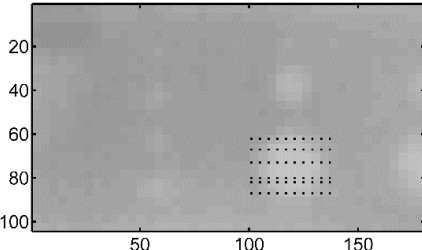

**Figure 11.** Segments of consecutive frames (broken lines) selected for intensity variation analysis.

The temperature variation over time is apparent, but so are the effects of lateral diffusion (as in the case of line 82 in Figure 10, top). In the specific case, the central part that corresponded to the 120th pixel exhibited a lower intensity than the pixels to the immediate left and right. This clearly shows that after initial heating, upon irradiation of the test sample, the heat started to spread from the center to the periphery of the defect.

The same figure and the intensity variation along line 73 (Figure 9, top) indicate that there was alternating heating and cooling of certain parts, manifested by changes in intensity of certain pixels due to heat diffusion in the defect area, immediately after the cooling process began.

Figures 5 and 6 show a non-uniform level of gray within the zone of the defect with the largest radius in column B (Procedure B, parameter *k* = 6, 9, 12 and 15) and column C (Procedure B, parameter *k* = 12, 15, 18, 21 and 24). Therefore, this effect is attributed to lateral diffusion. Its existence is apparent and depends on the circumference and depth of the defect at certain frequencies. For example, the effect is not visible in Figure 6 at *k* = 24, in the case of the defect with the largest radius in column B. It should be kept in mind that the sampling frequency is then low enough, so the effect that results from the transient process in the form of lateral diffusion is not apparent.

Figures 12–14 show the development of the thermal process in the defect areas along verticals A and B. The curves represent the change in the mathematical expectation of pixel intensity within squares whose dimensions correspond to the defect diameters ($16 \times 16$ and $8 \times 8$ pixels for $\emptyset$ = 15 and 8 mm, and 6x6 pixels for $\emptyset$ = 5 and 3 mm), relative to the mathematical expectation of pixel intensity in Sa (Figure 2). The centers of these areas coincide with the centers of the corresponding defects. The diameter of the cylindrical defects along vertical A ($\emptyset$ = 5 mm) was selected to test the nature of the transition process, which takes place after irradiation of the test sample, when the diameter was comparable to the depth. The curve maxima shown in Figures 12 and 13 occurred as expected, meaning that their value declined with increasing defect depth. However, local minima were noted immediately after the temperature rose due to irradiation and became increasingly pronounced when the defect depth was comparable to the diameter (curves b and c in Figure 12 and curves in Figure 13). This phenomenon attested to the existence of lateral heat diffusion towards the defect-free area of the test sample. Then a reverse diffusion process began, or the heat started to spread again, toward the central area above the defects, and they were re-heated. This phenomenon was not interesting only in terms of physics. It allowed greater insight into the reason why detection was more difficult when the diameter was small ($\emptyset$ = 3 mm defect was not detected along verticals D and E).

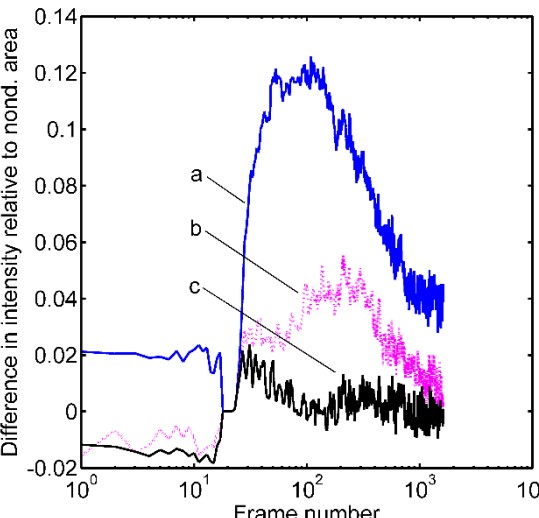

**Figure 12.** Change in mathematical expectation of pixel intensity in ∅ = 5 mm defect area along vertical A at depths: (a) *h* = 3.7 mm, (b) *h* = 3.4 mm and (c) *h* = 2.7 mm.

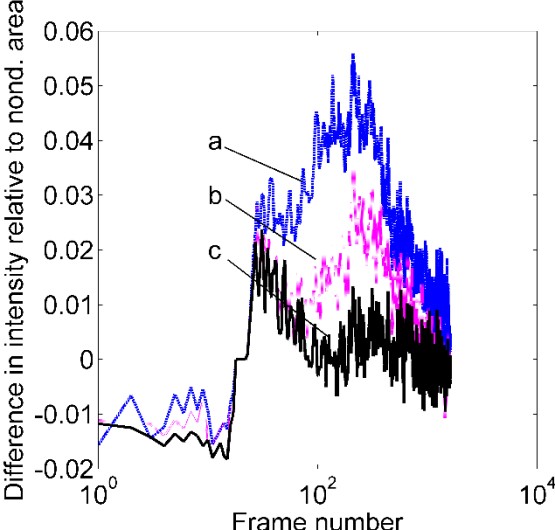

**Figure 13.** Change in mathematical expectation of pixel intensity in ∅ = 5 mm defect area along vertical A at depths: (a) *h* = 3.4 mm, (b) *h* = 3.2 mm and (c) *h* = 2.7 mm.

Figure 14 depicts the effect of reducing the diameter of the defects at the same depth. The results shown relate to the defects along vertical B. The highest maximum value, as expected, was noted in the case of the largest-diameter defect (curve a). Keeping in mind that the depth of all the defects along vertical B was the same and that diameter ∅ = 15 mm was the largest, it was reasonable to expect that the transition process would be the slowest in that case and for this maximum to occur last, or after the maxima of curves b and c relating to defect diameters ∅ = 8 mm and ∅ = 3 mm.

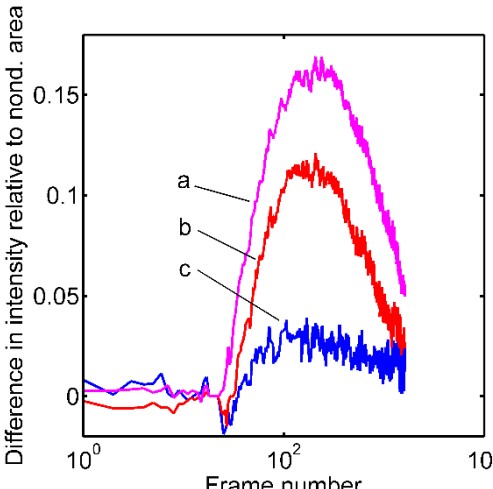

**Figure 14.** Change in mathematical expectation of pixel intensity in defects areas along vertical B: (a) ∅ = 15 mm, (b) ∅ = 8 mm, and (c) ∅ = 3 mm.

## 4. Conclusions

A comparative analysis of experimental results processed applying two procedures in the frequency domain indicated that the processing approach based on the introduction of zero frames provided better insight into the shape and position of the defects. After they were introduced, the frequency of appearing frames that took part with their pixel values in the computation of spectral components corresponded to the desired sampling frequency. At the same time, the number of samples increased and resulted in a better resolution in the frequency domain. Additional analyses in the time domain corroborated the results derived from the frequency domain.

**Author Contributions:** L.T., designed and performed the experiments. V.D., analyzed the experimental data and wrote the paper. G.D., processed the experimental data and produced the graphics. B.M., provided advice on the discussion of results.

**Funding:** This work was done within the research project of the Ministry of Science and Technological Development of Serbia III47029.

**Conflicts of Interest:** The authors declare no conflict of interest.

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
