# Peer review of "Reconstruction of Simulated Cylindrical Defects in Acrylic Glass Plate Using Pulsed Phase Thermography"

_applsci, doi:10.3390/app9091854_

Round 1

Reviewer 1 Report

It is not clear, what's new. The content is the state of 

the art.

What is the scientific gain?

Author Response

Response to Reviewer 1 Comments

Point 1

It is well known that defect depth can be estimated by processing a sequence of thermal images captured at different sampling frequencies. However, the question arises how to attain, through data processing, the effects that would be achieved at lower sampling frequencies, if the lowest sampling frequency of a specific camera is higher than required.

The starting point was a simple idea (Procedure A) to simulate the effect of recording at a lower sampling frequency by uniform frame extraction from the available sequence of thermal images captured at a higher sampling frequency. This resulted in fewer frames and, upon FFT processing, a smaller number of spectral components, which can reduce the visibility of defects in the tested material.

For that reason, a new procedure (Procedure B) is proposed, in which an appropriate window function in the time domain is used. Then FFT processing ensures that the frames extracted in the previous procedure are also used and the effect of all the other frames (zero frames) is disregarded.

Point 2

The introduction of zero frames, using the window function in the time domain, attains the effect of data processing achievable with a sampling frequency harmonized with defect depth. 

Reviewer 2 Report

The current research work presents the application of PPT for the detection of simulated defects in an acrylic glass plate.

The motivation behind this study is absolutely interesting, nevertheless prior its publication some  modifications shall be done which I believe if they will be addressed, they will enhance the quality of the paper. My comments are:

Authors are strongly advised to describe the objective of this work in the introduction part. PPT has been widely used for the detection/quantification of internal defects and several published works have demonstrated the importance of the sampling frequency in similar studies. Nonetheless,from the current manuscript version the objective behind the selection of these two procedures and their added value to the already published work in PPT are not clear.

More information should be included at the description of Procedures A & B. For instance why 64 frames were extracted in the former one or what was the aim of using zero frames in the latter procedure?

Indeed from Figs. 5 and 6, it seems that that procedure B can provide results with enhanced resolution.Considering that your main objective is to compare the applicability of these two procedures, the discussion of the results coming from the time domain analysis should be linked to the aforesaid.

Finally, the conclusion part should be re-structured, authors are strongly advised to describe and summarise the key findings from this study and not a summarised description of the work carried out.

Author Response

Response to Reviewer 2 Comments

Point 1

It is well known that defect depth can be estimated by processing a sequence of thermal images captured at different sampling frequencies. However, the question arises how to attain, through data processing, the effects that would be achieved at lower sampling frequencies, if the lowest sampling frequency of a specific camera is higher than required.

The starting point was a simple idea (Procedure A) to simulate the effect of recording at a lower sampling frequency by uniform frame extraction from the available sequence of thermal images captured at a higher sampling frequency. This resulted in fewer frames and, upon FFT processing, a smaller number of spectral components, which can reduce the visibility of defects in the tested material.

For that reason, a new procedure (Procedure B) is proposed, in which an appropriate window function in the time domain is used. Then FFT processing ensures that the frames extracted in the previous procedure are also used and the effect of all the other frames (zero frames) is disregarded.

Thus, the proposed solution improves the resolution and detection of defects at different depths, if the camera does not support the required operating mode.

Point 2

The text below in red was added to the description of Procedures A and B:

In the first procedure (Procedure A), 64 frames were extracted in the order: k= 3, 6, 9, 12, 15, 18, 21 and 24, using the loop FOR i= k0 : k : (k0+k×63). This resulted in the minimum sampling frequency of fmin= 30/24 samples per second. For each pixel selected in this manner with 64 frames, vector x(k) was generated and used in the DFT described by Eq. (1). A total of 1617 frames were captured during the experiment. Keeping in mind that at k=24 at least 1+24×63= 1513 are required, any further frequency reduction was not possible. On the other hand, if only 32 frames were used, a smaller number of spectral components would be obtained (i.e. the resolution in the frequency domain would be poorer).

To enhance resolution, in the second procedure (Procedure B) the number of samples N was increased, but without adding information by selecting additional frames from the basic sequence of all recorded frames. Instead, “zero frames” were introduced (i.e. the values of all pixels in the corresponding matrix were equal to zero). This meant that a sample vector was generated each time and its every k-th term, beginning with the first, had the value of the corresponding pixel within the sequence of the 64 frames selected from the basic sequence, as in the case of Procedure A. All other terms were zero. The total number of terms N of the vector generated in this way was equal to the minimum value of the power of 2, which ensured that the pixel values of all frames extracted from the basic sequence by means of the previously-mentioned FOR NEXT loop were stored.

It follows from Eq. (1) that all samples of signal x(k), recorded in the time domain, participate in the computation of all spectral components X(n) in the frequency domain. The introduction of zero frames had the indirect effect of sampling frequency reduction. Namely, the frequency of appearing frames that participate with their pixel values in the computation of spectral components X(n) corresponds to the desirable sampling frequency. For example, at k=3 the frequency of appearing frames whose elements differed from zero was lower by a factor of three.

Point 3

The discussion of the results in the time domain was linked with the results in the frequency domain by adding two paragraphs in Section 3. Results and discussion:

 It is apparent that the time lag of the peak of the corresponding curve, in Fig. 4, is the longest in the case of the defects along vertical E. Namely, it corresponds to the defect that is most distant from the heated surface. Obviously, there is a correlation with the results shown in Figs. 4 and 5. As the distance of the defects (of the same radii) increases, viewed relative to the heated surface, so does the time lag. This is consistent with the fact that a longer sampling period, or a lower blind frequency, is required for them to be visible.

………….

………….

……………

Figures 5 and 6 show a non-uniform level of gray within the zone of the defect with the largest radius in column B (Procedure B, parameter k = 6, 9, 12 and 15) and column C (Procedure B, parameter k = 12, 15, 18, 21 and 24). Therefore, this effect is attributed to lateral diffusion. Its existence is apparent and depends on the circumference and depth of the defect at certain frequencies. For example, the effect is not visible in Fig. 6 at k = 24, in the case of the defect with the largest radius in column B. It should be kept in mind that the sampling frequency is then low enough, so the effect that results from the transient process in the form of lateral diffusion is not apparent.

Point 4

Conclusion was revised in its entirety as suggested (key results are described and summarized).

Round 2

Reviewer 2 Report

The present improved version of the manuscript can be considered for publication. I have no other comment for the authors